# Duplex One-Step RT-qPCR Assays for Simultaneous Detection of Genomic and Subgenomic RNAs of SARS-CoV-2 Variants

**DOI:** 10.3390/v14051066

**Published:** 2022-05-17

**Authors:** Sushma M. Bhosle, Julie P. Tran, Shuiqing Yu, Jillian Geiger, Jennifer D. Jackson, Ian Crozier, Anya Crane, Jiro Wada, Travis K. Warren, Jens H. Kuhn, Gabriella Worwa

**Affiliations:** 1Integrated Research Facility at Fort Detrick, National Institute of Allergy and Infectious Diseases, National Institutes of Health, Frederick, MD 21702, USA; sushma.bhosle@nih.gov (S.M.B.); julie.tran@nih.gov (J.P.T.); shuiqing.yu@nih.gov (S.Y.); jillian.geiger@nih.gov (J.G.); Jennifer6814@gmail.com (J.D.J.); anya.crane@nih.gov (A.C.); wadaj@mail.nih.gov (J.W.); travis.warren@nih.gov (T.K.W.); 2Clinical Monitoring Research Program Directorate, Frederick National Laboratory for Cancer Research, Frederick, MD 21701, USA; ian.crozier@nih.gov

**Keywords:** SARS-CoV-2, variant, duplex RT-qPCR, RNA, genomic, subgenomic

## Abstract

A hallmark of severe acute respiratory syndrome virus (SARS-CoV-2) replication is the discontinuous transcription of open reading frames (ORFs) encoding structural virus proteins. Real-time reverse transcription PCR (RT-qPCR) assays in previous publications used either single or multiplex assays for SARS-CoV-2 genomic RNA detection and a singleplex approach for subgenomic RNA detection. Although multiplex approaches often target multiple genomic RNA segments, an assay that concurrently detects genomic and subgenomic targets has been lacking. To bridge this gap, we developed two duplex one-step RT-qPCR assays that detect SARS-CoV-2 genomic ORF1a and either subgenomic spike or subgenomic ORF3a RNAs. All primers and probes for our assays were designed to bind to variants of SARS-CoV-2. In this study, our assays successfully detected SARS-CoV-2 Washington strain and delta variant isolates at various time points during the course of live virus infection in vitro. The ability to quantify subgenomic SARS-CoV-2 RNA is important, as it may indicate the presence of active replication, particularly in samples collected longitudinally. Furthermore, specific detection of genomic and subgenomic RNAs simultaneously in a single reaction increases assay efficiency, potentially leading to expedited lucidity about viral replication and pathogenesis of any variant of SARS-CoV-2.

## 1. Introduction

The ongoing coronavirus disease 2019 (COVID-19) [1] pandemic [2] is caused by severe acute respiratory syndrome coronavirus 2 (SARS-CoV-2), a sarbecovirus of the pisoniviricete order *Nidovirales* [3,4]. Since the beginning of the pandemic in late 2019 [5,6], SARS-CoV-2 has evolved into numerous distinct lineages, including variants of concern with varying levels of transmission, immune evasion, and disease severity [7].

SARS-CoV-2 has a capped, polyadenylated, linear, nonsegmented, positive-sense RNA genome that is ≈30 kb in length. The SARS-CoV-2 genome begins with a 5′ leader and a transcriptional regulatory sequence (TRS), followed by two overlapping open reading frames (ORFs), designated as *ORF1a* and *ORF1b*, that comprise about two-thirds of the entire genome. Two polyproteins, generated during direct translation of genomic *ORF1a* (*gORF1a*) and or *ORF1a* merged with *ORF1b* via programmed ribosomal frameshifting (pp1ab), are processed via co-translational and post-translational mechanisms to yield at least 16 nonstructural proteins that mediate, among other functions, genome replication and transcription. The remaining one-third of the genome, which includes the 3′ end, contains numerous ORFs that encode at least four structural proteins, spike (S), membrane (M), envelope (E), and nucleocapsid (N), and eight accessory proteins, 3a, 3b, 6, 7a, 7b, 8b, 9b, and 14 [8,9]. These ORFs are transcribed into a nested set of negative-sense subgenomic RNAs (sgRNAs) that in turn serve as templates to create a nested set of mRNAs. Integral to this process are TRSs adjacent to each ORF [10,11].

Many real-time reverse transcription PCR (RT-qPCR) assays and multiplexing kits (including triplex and quadruplex approaches) have been developed for SARS-CoV-2 detection [12]. A singleplex one-step RT-qPCR assay, developed by the U.S. Centers for Disease Control and Prevention (CDC) for the detection of two different regions of SARS-CoV-2 genomic *N* (*gN*) [13] targets, is widely used [14]. Several other one-step RT-qPCR assays targeting *gN* have been adapted for SARS-CoV-2 but were found to be less sensitive than the widely used CDC assay [15]. One multiplex one-step RT-qPCR assay was able to simultaneously detect two different regions of the *gN* sequence of SARS-CoV-2 using automated analysis [16]. Several duplex RT-qPCR assays were optimized for simultaneous detection of genomic targets, such as *gN* or genomic *E* (*gE*), with a human cellular gene control [17,18]. A triplex RT-qPCR assay simultaneously detected either of two different regions of *gN* [19,20] or *N* and *E* target genes with a human cellular gene target [21,22]. Moreover, a quadruplex RT-qPCR assay has been used for the detection of *gN* and *gE* along with two human cellular genes [23]. Some triplex RT-qPCR assays also target the genomic *S* (*gS*) [7] and *gE* or *S* and *ORF8* genes simultaneously with human cellular genes [24,25]. Additionally, some multiplex RT-qPCR assays simultaneously detect two different nidovirals, porcine torovirus (PToV) and coronavirid [26]. Furthermore, the CDC and other organizations have optimized multiplex RT-qPCR assays for diagnostic detection of influenza A and B viruses and SARS-CoV-2 in single reactions [27,28,29]. Multiplex RT-qPCR assays that target variant-specific genomic sequences of *S* and *ORF1a* also make it possible to differentiate SARS-CoV-2 variants from the original strain isolated in Wuhan [30].

Several reverse transcription droplet digital PCR (RT-ddPCR) assays [13,31] have been developed to detect sgRNA with high precision. However, sgRNA targets have not yet been incorporated into multiplex RT-qPCR assays because the repetition of ORFs in sgRNA transcripts and skipping only one ORF in each subsequent transcript during the discontinuous transcription process, which could interfere with the specific detection of sgRNA. When detecting the SARS-CoV-2 *gN* target, assays cannot distinguish an actively replicating virus from the input viral load; however, RT-qPCR detection of subgenomic *E* (*sgE*) differentiates an actively replicating virus from the viral RNA load [32]. Furthermore, fewer sgRNAs were found in COVID-19 patients and SARS-CoV-2-exposed animals after treatment with antivirals, suggesting that sgRNA concentration may help track the success of the therapeutic intervention [1,32]. On the contrary, some studies reported high correlation and similar trends between genomic and subgenomic RNA copy numbers longitudinally in post-exposure clinical samples, indicating a limited additional benefit of detecting sgRNA [33,34].

Building on existing RT-qPCR assays, we developed two duplex one-step assays that detect *gORF1a* and either subgenomic *S* (*sgS*) or subgenomic *ORF3a* (*sgORF3a*) targets. We selected *sgS/sgORF3a* for use with *gORF1a* because they are proximally transcribed as the first two sgORFs in the series of discontinuous transcripts and *sgS* is the longest of the non-1a/1b ORFs; thus, selecting these targets could potentially eliminate the chances of detecting nonspecific ORFs. Moreover, detection of *sgS* concentration is important as mutations in the RNA binding domain of *sgS* effect membrane fusion and viral entry, thus influencing the transmission of virus [35,36]. Detection of both *sgS* and *sgORF3a* are important as they are involved in cytokine release, immune dysfunction and have a major role in pathogenesis [37]. We applied our duplex RT-qPCR assays to improve our understanding of the replication kinetics of the Washington strain and delta variant of SARS-CoV-2.

## 2. Materials and Methods

### 2.1. Cells

Grivet (*Chlorocebus aethiops* (Linnaeus, 1758)) kidney epithelial Vero E6 cells expressing human transmembrane serine protease 2 (TMPRSS2 Vero E6; Japanese Collection of Research Bioresources [JCRB] Cell Bank, Osaka, Japan; JCRB1819) were cultured in Gibco low-glucose modified Dulbecco’s Modified Eagle’s Medium (DMEM; Thermo Fisher Scientific, Waltham, MA, USA) containing 10% fetal bovine serum (FBS; MilliporeSigma, Burlington, MA, USA) and 1 mg/mL of G 418 disulfate salt solution (MilliporeSigma), and Grivet Vero cells (American Type Culture Collection [ATCC], Manassas, VA, USA; #CCL-81) were cultured in Gibco minimum essential medium (MEM; Thermo Fisher Scientific) containing 2% heat-inactivated FBS (MilliporeSigma). Human lung epithelial Calu-3 cells (ATCC #HTB-55) were cultured in Gibco Roswell Park Memorial Institute medium (RPMI) 1640 Medium (Thermo Fisher Scientific) with 10% FBS. All cells were maintained in a humidified incubator at 37 °C with 5% carbon dioxide (CO_2_).

### 2.2. Viruses

Passage 3 of severe acute respiratory syndrome coronavirus 2 (SARS-CoV-2; *Coronaviridae: Sarbecovirus*) Washington strain (SARS-CoV-2/human/USA/WA-CDC-02982586-001/2020; GenBank #MN985325) was obtained from the CDC, Atlanta, GA, USA. SARS-CoV-2 Washington strain was passaged once at the Integrated Research Facility at Fort Detrick (IRF-Frederick) in Vero cells for 72 h. The resulting virus stock (SARS-CoV-2/human/USA/WA-CDC-WA1/2020 [IRF-0394]; GenBank #MW161259) was titrated by plaque assay using Vero E6 cells (ATCC #CRL-1586) cultured in DMEM media with 10% FBS, as previously described [38]. Subsequently, under identical growth conditions and media, Vero cells were exposed to the IRF-0394 virus stock at a multiplicity of infection (MOI) of 0.01 to generate working stocks (SARS-CoV-2/human/USA/WA-CDC-WA1/2020 [IRF-0399]; GenBank #MT952134 and SARS-CoV-2/human/USA/WA-CDC-WA1/2020 [IRF-0449]; GenBank #OK091601). Cells were incubated with either working stock IRF-0399 or IRF-0449 for 48 h, and media were collected after 3 freeze/thaw cycles. Media were clarified via centrifugation at 7500× *g* for 10 min.

SARS-CoV-2 delta variant (SARS-CoV-2/human/USA/PHC658/2021) was obtained from BEI Resources, Manassas, VA, USA (#NR-55611). SARS-CoV-2 delta variant was passaged at the IRF-Frederick in Calu-3 cells. Supernatant was collected at 48 h after 3 freeze/thaw cycles. The resulting virus stock (SARS-CoV-2/human/USA/PHC658/2021 [IRF-0455]; GenBank #OM673006) was clarified at 7500× *g* for 10 min. All virus stocks were quality-controlled as previously described [39].

### 2.3. RNA Extractions

RNA extracted from IRF-0399 cultures was used for developing and optimizing singleplex and duplex RT-qPCR assays. The RNA extraction method was based on earlier published methods with some modifications [40]. Briefly, a volume of 0.75 mL of TRIzol LS reagent (Thermo Fisher Scientific) was added to 0.25 mL of Vero cells supernatants, and samples were transferred to Invitrogen Phasemaker Tubes (Thermo Fisher Scientific). Samples were incubated for 5 min and mixed with 0.2 mL of chloroform (MilliporeSigma) per 1 mL of TRIzol LS. Samples were then mixed vigorously by shaking, incubated for 10–15 min, and centrifuged at 16,000× *g* for 5 min at 4 °C. Then, 0.5 mL of isopropanol (MilliporeSigma) were added to the aqueous phase per 1 mL of TRIzol LS. Samples were incubated for 10 min and centrifuged at 12,000× *g* for 10 min at 4 °C. RNA pellets were resuspended in 1 mL of 75% ethanol (Thermo Fisher Scientific) per 1 mL of TRIzol LS. Samples were centrifuged at 7500× *g* for 5 min at 4 °C. RNA pellets were then air-dried and resuspended in 30 µL of RNase-free water (Integrated DNA Technologies [IDT], Coralville, IA, USA).

Cultures from IRF-0449 and IRF-0455 were used for the time-course experiments, and RNA was extracted from infected TMPRSS2 Vero E6 cells at various time points. RNA extraction from TMPRSS2 Vero E6 cells was performed with the MagMAX Viral Pathogen Nucleic Acid Isolation (MVP I) kit (Thermo Fisher Scientific) and KingFisher Flex Purification System on TRIzol-LS-lysed cells according to the manufacturer’s instructions and with some modification from the method earlier described [41]. Briefly, in a template-free hood, using the KingFisher deep 96-well plates (Thermo Fisher Scientific), 1000 µL of wash buffer from the MVP I kit was aliquoted in Plate 1, 1000 µL of 80% ethanol in Plate 2, 500 µL of 80% ethanol in Plate 3, and 70 µL elution buffer from the MVP I kit in Plate 4. The sample plate was prepared by adding 10 µL of proteinase K (Thermo Fisher Scientific) and 200 µL of TRIzol-LS-lysed cells. A volume of 550 µL binding mixture beads was added to the sample plate (KingFisher deep 96-well plate), ensuring that each well was filled. The plates were loaded onto the KingFisher Flex Purification System along with a deep 96-well tip comb (Thermo Fisher Scientific), and RNA extraction was performed using the MVP_Flex_200µL.bdz run program (Thermo Fisher Scientific). All plates were removed from the system after the run was completed. The elution plate was placed onto a plate stand with a magnetic base to remove any residual magnetic beads. The eluted RNA samples were collected and stored at −80 °C. The RNA samples were quantified with an Invitrogen Qubit RNA quantification high sensitivity kit (Thermo Fisher Scientific), and 10 pg of RNA was set aside for use in the RT-qPCR duplex assay.

### 2.4. Gene Fragment Synthesis

All gene fragments (gBlocks) were supplied by IDT. A gene fragment containing sequences from SARS-CoV-2 gORF1a, sgS, and sgORF3a was used as a positive control, and a gene fragment specific for gS and gORF3a was used as a negative control. The gene fragment design was based on SARS-CoV-2 Wuhan-Hu-1 isolate (GenBank #MN908947).

### 2.5. RT-qPCR Assays

TaqPath 1-Step RT-qPCR Master Mix (Thermo Fisher Scientific) was used to perform singleplex and multiplex RT-qPCR assays. The forward primer sequence to detect sgRNAs was the leader sequence [42], and the primers and probe for *ORF1a* were from published work [43]. All other primers and probes were designed using IDT primer design software to achieve optimal annealing temperature, length, percent guanine–cytosine (GC) content, delta G, and amplicon size. Primer pairs and probe design with fluorophore and quenchers are shown in Table 1. A master mixture containing thermostable Moloney murine leukemia virus (MMLV) reverse transcriptase, deoxynucleoside triphosphates (dNTPs), uracil-DNA N-glycosylase (UNG), carboxy-X-rhodamine (ROX) dye, thermostable Fast DNA polymerase reagent (12 μL), and 3 μL of primers and probes (250 nM) was prepared. A volume of 5 μL of template was added to 15 μL of master mix for each reaction. Mixtures were subjected to complementary DNA (cDNA) synthesis (2 min at 25 °C, followed by 15 min at 55 °C), initial denaturation (2 min at 95 °C), 40 cycles of subsequent denaturation (each 10 s at 95 °C), annealing/elongation (30 s at 58 °C), and final elongation (2 min at 72 °C)—followed by holding at 4 °C in a QuantStudio 7 Flex Real-Time PCR System (Thermo Fisher Scientific). RNase-free water was used as the non-template control. RT-qPCR amplicons were analyzed on 2% agarose gels (Thermo Fisher Scientific). The sequence of gene fragments tested for specificity of the assays are shown in Table 1.

### 2.6. Time-Course Experiments

TMPRSS2 Vero E6 cell monolayers in Corning 6-well plates (Thermo Fisher Scientific) were inoculated with 0.01 MOI of SARS-CoV-2 Washington strain or delta variant for 1 h at 37 °C with 5% CO_2_. Cells were rinsed once with 1 mL of Dulbecco’s phosphate buffered saline (DPBS), no calcium, no magnesium (Gibco), and 3 mL of fresh DMEM were added to the cells. The cells were cultured in an incubator at 37 °C; supernatants and cells were harvested every 2 h for the first 8 h, every 4 h up to 24 h, and every 6 h up to 48 h post-exposure. Cells were visualized at 10× magnification, and the cytopathic effect was observed at each time point. Supernatants from each well were placed into separate 15-mL conical tubes (Crystalgen, Commack, NY, USA) and centrifuged at 1400× *g* for 10 min at 4 °C. A total of 1 mL of supernatant from each tube was aliquoted and stored at −80 °C for plaque assay. Cells were rinsed with 3 mL of DPBS to remove residual supernatant, and 1 mL of media was added to each well. Cells were scraped off the plates, aliquoted, and inactivated with TRIzol LS for RT-qPCR assays. Supernatants collected from infected cells at multiple time points were used for plaque assays and duplex RT-qPCR assays, respectively. Duplex RT-qPCR assays for *gORF1a + sgS* and *gORF1a + sgORF3a* in cells were performed for the SARS-CoV-2 Washington strain and the delta variant for all time points. Results were analyzed for Ct values with a threshold setting of 0.08 for *gORF1a + sgS* assay and 0.04 for *gORF1a + sgORF3a* and plotted for each time point and variant.

Plaque assays were performed with 6 dilutions of each sample from 10^−1^–10^−6^ in Gibco low-glucose DMEM with GlutaMAX (Thermo Fisher Scientific). TMPRSS2 Vero E6 cells were plated in 6-well plates, and media was aspirated the next day when cells reached 90–100% confluency. Cells were exposed to 330 µL of the serially diluted supernatant from each sample in duplicate. Cells were incubated for 60–70 min at 37 °C, rocking gently approximately every 10–15 min. A 2-mL Avicel RC-591 overlay (FMC BioPolymer, Philadelphia, PA, USA) and 2× of plaque assay media at 1:1 ratio was added to each well. Plates were swirled gently to mix, covered, and then incubated for 34 h at 37 °C. Plates were then swirled to loosen the overlay, and the overlay was removed carefully. A volume of 1–2 mL of 0.2% crystal violet (RICCA, Arlington, TX, USA) was added to each well. Plates were swirled gently to mix, covered, and then incubated for at least 30 min at ambient temperature. Subsequently, crystal violet was aspirated, and plates were rinsed thoroughly with tap water and dried at ambient temperature. Plaques were then counted on a lightbox. Viral titers were calculated as plaque-forming units (PFU) per mL [38].

### 2.7. Statistics

All analyses were performed using Prism version 9.1.1.225 (GraphPad Software, San Diego, CA, USA). The standard curves for singleplex and duplex assays were attained with linear regression analysis; the linear equations and R^2^ values are shown in Figure 1 and Figure 2. Temporal changes in genomic and subgenomic RNA copies in the time-course experiment were analyzed by two-way ANOVA with Bonferroni’s multiple comparisons test with a 99.9% confidence interval. Results with difference in cycle threshold (Ct) values with *p* < 0.01 were considered significant. The difference of SARS-CoV-2 Washington strain and delta variant titers at various time points were analyzed with a parametric unpaired two-tailed t-test with a 95% confidence interval, and *p* < 0.05 was considered significant.

## 3. Results

### 3.1. Singleplex RT-qPCR Assays Were Established and Optimized for the Detection of SARS-CoV-2 Genomic and Subgenomic Nucleic Acids Using Synthetic Gene Fragments

To develop a duplex RT-qPCR assay for detection of SARS-CoV-2 genomic and subgenomic nucleic acids, we first established and optimized singleplex RT-qPCR assays targeting *gORF1a*, *sgS*, or *sgORF3a* sequences. Synthetic gene fragments, along with primers and probes (Figure 3, Appendix A and Table 1), were used to generate standard curves for each target region. *gORF1a* was targeted with primers and probes specific to *gORF1a* (Appendix A). To avoid detection of genomic RNAs in subgenomic RT-qPCR assays, we used the forward primers to target sgRNA leader sequences as these are not present in genomic sequences (Appendix A). Standard curves were generated in triplicate using synthetic gene fragments, serially diluted by 10-fold from 1 × 10^8^ copies per µL to 10 copies per µL. The slope for the *gORF1a* singleplex RT-qPCR assay was −3.437, with a correlation coefficient of R^2^ = 0.9999. The slopes for the *sgS* and *sgORF3a* singleplex RT-qPCR assays were −3.459 and −3.355, respectively. The correlation coefficients were R^2^ = 0.9998 for *sgS* and R^2^ = 0.9993 for *sgORF3a*. The Ct values expressed on a linear regression plot showed a difference of 3–4 with each log_10_ dilution (Figure 1A–C). The amplification plots for the standard curves generated for *gORF1a, sgS,* and *sgORF3a* singleplex RT-qPCR assays are shown in Appendix A. These results indicated the detection of *gORF1a*, *sgS*, and *sgORF3a* using synthetic gene fragments for standard curve quantification.

### 3.2. Singleplex RT-qPCR Assays Specifically Detected SARS-CoV-2 Genomic or Subgenomic Nucleic Acids

We extracted RNA from the supernatant of Washington-strain-infected Vero cells at 48 h post-inoculation. Isolated RNA was used as a template for *gORF1a*, *sgS*, and *sgORF3a* singleplex assays. Specifically designed and synthesized gene fragments for *gORF1a*, *sgS*, and *sgORF3a* were used as positive controls, while a non-template control (NTC) and *gS* and *gORF3* synthetic gene fragments lacking a leader sequence served as negative controls.

Prior to performing the RT-qPCR, we evaluated each primer pair individually with a singleplex qualitative reverse transcription PCR (RT-PCR) assay, which does not use fluorescent probes. Rather, amplification products in positive controls and RNA samples were visualized on a gel, and no amplification was observed in the negative controls (Appendix A). The specificity of each singleplex assay was determined by using appropriate positive and negative synthetic gene fragment controls.

To evaluate whether the designed primers and probes specifically detect the respective SARS-CoV-2 RNAs, we used the template for *gORF1a*, *sgS*, and *sgORF3a*, along with synthetic gene fragments as positive controls and an NTC and synthetic gene fragments as negative controls. Single bands were visible on agarose gels corresponding to *gORF1a* (118 bases), *sgS* (201 bases), and *sgORF3a* (289 bases), respectively, in SARS-CoV-2 RNA and the respective positive control reactions—but not in negative control reactions. As expected, template amplification was confirmed in SARS-CoV-2 RNA and positive control samples (Figure 4A–C). Ct values for all singleplex RT-qPCR assays are listed in Table 2.

### 3.3. Duplex RT-qPCR Assays Specifically Detected SARS-CoV-2 Genomic and Subgenomic Nucleic Acids

After confirming detection using the *gORF1a*, *sgS*, and *sgORF3a* singleplex RT-qPCR assays, we established duplex RT-qPCR assays capable of detecting SARS-CoV-2 *gORF1a* and *sgS* or *sgORF3a* simultaneously in single reactions. Standard curves were generated by mixing the respective synthetic gene fragments of *gORF1a* or *sgS* (1 × 10^8^ copies per µL) and *gORF1a* or *sgORF3a* fragments (1 × 10^7^ copies per µL) 10-fold serially until 10 copies per µL were achieved. Figure 2 shows the linear regression curves. The slopes for the *gORF1a + sgS* duplex assay were −3.280 for *gORF1a* and −3.339 for *sgS* (Figure 2A); the slopes for the *gORF1a + sgORF3a* duplex assay were −3.383 for *gORF1a* and −3.384 for *sgORF3a* (Figure 2B). The correlation coefficients for the *gORF1a + sgS* duplex assay were R^2^ = 0.9997 for both *gORF1a* and *sgS* (Figure 2A), and the correlation coefficients for the *gORF1 + sORF3a* duplex assay were R^2^ = 0.9996 for *gORF1a* and R^2^ = 0.9997 for *sgORF3a* (Figure 2B). The amplification plots for the standard curves generated for the two duplex assays showed proportionate increases in Ct values with each log_10_ dilution (Appendix A).

Next, we evaluated the duplex assays using RNA isolated from supernatants of Vero cell cultures inoculated with SARS-CoV-2 Washington strain, as described above. An equal mixture of gene fragments for *gORF1a* or *sgS* and *gORF1a* or *sgORF3a* at the concentration of 1 × 10^8^ copies per µL was used as a positive control. For *gORF1a + sgS* and *gORF1a + ORF3a* duplex assays, amplification was confirmed by using positive control gene fragments and SARS-CoV-2 RNA extracted from supernatants of infected Vero cells. Amplification products were separated on gels and revealed distinct bands following each duplex assay corresponding to the expected amplicon sizes (Figure 2C,D). The amplification plots for the *gORF1a + sgS* and *gORF1a + sgORF3a* reactions using SARS-CoV-2 RNA are shown in Appendix A. Using synthetic gene fragment standards, we achieved a detection limit of 50 copies per reaction of genomic and subgenomic RNA in the duplex assays; lowering the concentration to five copies did not successfully amplify the target RNA copies (Appendix A). These results established two standardized duplex one-step RT-qPCR assays for simultaneous detection of genomic and subgenomic RNAs in a single reaction.

### 3.4. SARS-CoV-2 Washington Strain and Delta Variant Genomic and Subgenomic RNAs Were Longitudinally Detected in Vero E6 Cells Expressing TMPRSS2

To evaluate whether the duplex RT-qPCR assays could detect different strains of SARS-CoV-2, we inoculated TMPRSS2 Vero E6 cells with either SARS-CoV-2 Washington strain or delta variant at an MOI of 0.01 and measured genomic and subgenomic viral RNA copies over time using the *gORF1a* + *sgS* and *gORF1a* + *sgORF3a* assays. The RNA copy numbers per µg of total RNA were calculated based on the respective synthetic gene fragments in parallel. As expected, genomic and subgenomic RNA copies increased considerably and similarly during the first 16 h after virus exposure independent of virus or assay (Figure 5). However, after 16 h, the Washington strain genomic and subgenomic RNA copy numbers reached a plateau (Figure 5A,C), whereas the delta variant genomic and subgenomic RNA copies continued to increase for an additional 4 h. Interestingly, only *gORF1a* and *sgORF3a* copies remained constant, whereas a marked reduction of *sgS* copies was observed after 30 h, followed by an increase (although not to the previously recorded maximum) by 42 h (Figure 5B,D). The Ct values and copy numbers per µg of total RNA for all experiments are shown in Appendix A. Summarized, these data suggest a similar abundance of *gORF1a* and *sgORF3a* in the supernatant of both variants over time but a markedly reduced concentration of *sgS* after 20 h of replication of the delta variant.

### 3.5. SARS-CoV-2 Washington Strain and Delta Variant Titers Correlated with Genomic and Subgenomic RNA Copy Numbers

The supernatant was collected every 4 h, starting at 8 h post-exposure (and every 6 h starting at 24 h) from TMPRSS2 Vero E6 cells infected with SARS-CoV-2, and viral titers were determined longitudinally; the last samples were collected at 48 h post-exposure. For the Washington strain, viral titer increased until 20 h and then remained constant, whereas, in the case of the delta variant, viral titer increased initially until 20 h and then declined at later time points (Figure 6A).

Next, we compared SARS-CoV-2 RNA copy numbers per µg of total RNA to SARS-CoV-2 titers at 20, 24, 30, and 36 h time points. Until the 16 h time point, *gORF1a*, *sgS*, and *sgORF3a* copies increased, corresponding with viral titers in the Washington strain growth curve (Figure 5A,C and Figure 6A). Similarly, up to the 16 h time point, *gORF1a*, *sgS*, and *sgORF3a* copies increased in parallel with plaque counts of the delta variant. Beyond 20 h, very few changes were observed in *gORF1a* and *sgORF3a* RNA concentrations, whereas viral titers declined. However, a significant decline in *sgS* copy numbers after 20 h was observed, which correlated with the reduction in titers beyond 20 h (Figure 5B,D and Figure 6A–C). Therefore, our data suggest that *sgS* could predict replication of the SARS-CoV-2 delta variant.

## 4. Discussion

Nidovirals, such as SARS-CoV-2, use discontinuous transcription of parts of their positive-sense RNA genomes to create a nested set of negative-sense sgRNAs that, in turn, serve as templates for transcription of mRNAs to encode virus structural and accessory proteins [10,42]. Assays available for detection of SARS-CoV-2 nucleic acids predominantly target genomic nucleic acids in either single or multiplex formats [15,17,18,21,44,45,46]. The main limitation of specifically detecting sgRNAs by RT-qPCR is unintended amplification of other shorter sgRNAs generated in the series of discontinuous transcripts. Specifically, binding of a primer pair to other sgRNA occurs due to a common leader forward primer and repeated occurrence of all ORFs in each subsequent sgRNA transcript, skipping only one ORF each time during the process of discontinuous transcription. Thus, the risk of amplifying adjacent unintended shorter sgRNA after the first two sgRNA (*sgS* and *sgORF3a*) is much higher. To overcome this issue, we attempted to detect either *sgS*, which is transcribed proximately without repetition in other sgRNAs and has the longest length, or *sgORF3a*, which is repeated only in *sgS* in the discontinuous series of subgenomic transcripts. Although *sgORF3a* is repeated in the *sgS* transcript, its distant location from the forward primer binding leader sequence region, due to the length of the *sgS* transcript, reduces the experimental chances of unintentionally amplifying *sgS* in the *sgORF3a* assay.

Here, we developed one-step singleplex RT-qPCR assays for the detection of *gORF1, sgS*, and *sgORF3a* targets using specific primers and probes. All primers and probes designed for our assays bind to variants of SARS-CoV-2, including the recently emerged omicron variant. A gene fragment containing SARS-CoV-2 *gORF1a* was used as a positive control for detecting *gORF1a.* In the *sgS* assay, gene fragments specific for *gS* (spanning the end of *ORF3a,* TRS, and *S*) and *sgS* (spanning the leader, TRS, and the *S* region) were designed for determining specificity for genomic and subgenomic regions, respectively. In the *ORF3a* assay, a gene fragment (spanning the leader, TRS, and the *ORF3a* region) was used as a positive control to show primer specificity for the *sgORF3a* assay. A gene fragment for the *ORF3a* region lacking the leader–TRS was used as a negative control. Further, we optimized two duplex assays for simultaneous *gORF1a + sgS* or *gORF1a + sgORF3a* detection. These duplex assays successfully detected and quantified SARS-CoV-2 in RNA extracted from the supernatants of TMPRSS2 Vero E6 cells infected with SARS-CoV-2 Washington strain or delta variant.

*sgORF3a* copy numbers were found close to *gORF1a* numbers in supernatants of SARS-CoV-2-exposed TMPRSS2 Vero E6 cells. This observation is consistent with earlier reports showing that SARS-CoV-2 mRNA copies in the cell-culture supernatants collected from infected TMPRSS2 Vero E6 cells were 100 times greater compared to those from Vero E6 cells and this effect is attributed to TMPRSS2, which functions as an enhancer of SARS-CoV-2 cell entry [47]. Further, we found a sound correlation of an increase in *gORF1a* and *sgS/sgORF3a* copies with an increase in SARS-CoV-2 Washington strain titers, indicating both as markers for replication. A decline in viral titers for the Washington strain at 42 h might be attributed to the cytopathic effect observed at later time points, as suggested by the observed drop in viral titers at 48 h for the Washington strain. In the case of the delta variant, the reduction in *sgS* copies tracked with the decline in viral titer after 20 h. The decline in *sgS* copies for the delta variant could be due to the mutations in the *S* ORF, as these mutations affect mRNA translation, leading to differential expression of the S protein [48], as well as a distinct structure and antigenicity of the S protein of the delta variant compared to other variants [49]. The possibility of host cell immune responses differentially inhibiting the transcription, translation of subgenomic mRNA, and generation of virions at time points beyond 20 h should be considered. Although longitudinal detection of genomic RNA is adequate for understanding replication of the Washington strain, the distinct pattern of *sgRNA* over the time post-exposure correlates with the growth of the delta variant, implicating *sgRNA* as a prominent marker for predicting the virus growth. Further exploration to understand the variant-specific differential *sgRNA* expression, replication pattern, and immunogenicity would be advantageous in designing novel treatment strategies for any variant of SARS-CoV-2, including omicron.

Other technically advanced approaches, such as, a CRISPR-Cas12-based method [50], next-generation sequencing [51], and RT-ddPCR assays [31] for SARS-CoV-2 RNA detection, would require availability of specialized instrumentation, and quantification of RNA copies would be expensive. The crucial limitation of duplex assays is the fact that they are not validated with clinical samples from SARS-CoV-2 patients. Our assay design is subject to the limitation that it detects only two select targets, not allowing for extrapolation of other sgRNA abundance in the sample. In contrast, a clear advantage of our assay is the simultaneous and specific quantification of genomic and subgenomic RNA in one step using an RT-qPCR platform that is widely used in laboratories.

## 5. Conclusions

Our pragmatic high-throughput approach can help reduce reagent costs, preserve precious sample volume, limit assay run time, and allow for easy protocol implementation. In conclusion, our duplex one-step RT-qPCR assay specifically targets genomic and subgenomic RNA simultaneously in a single reaction which may led to an increased understanding of viral replication and pathogenesis specific to any variant of SARS-CoV-2 or any contagious virus.

## Figures and Tables

**Figure 1 viruses-14-01066-f001:**
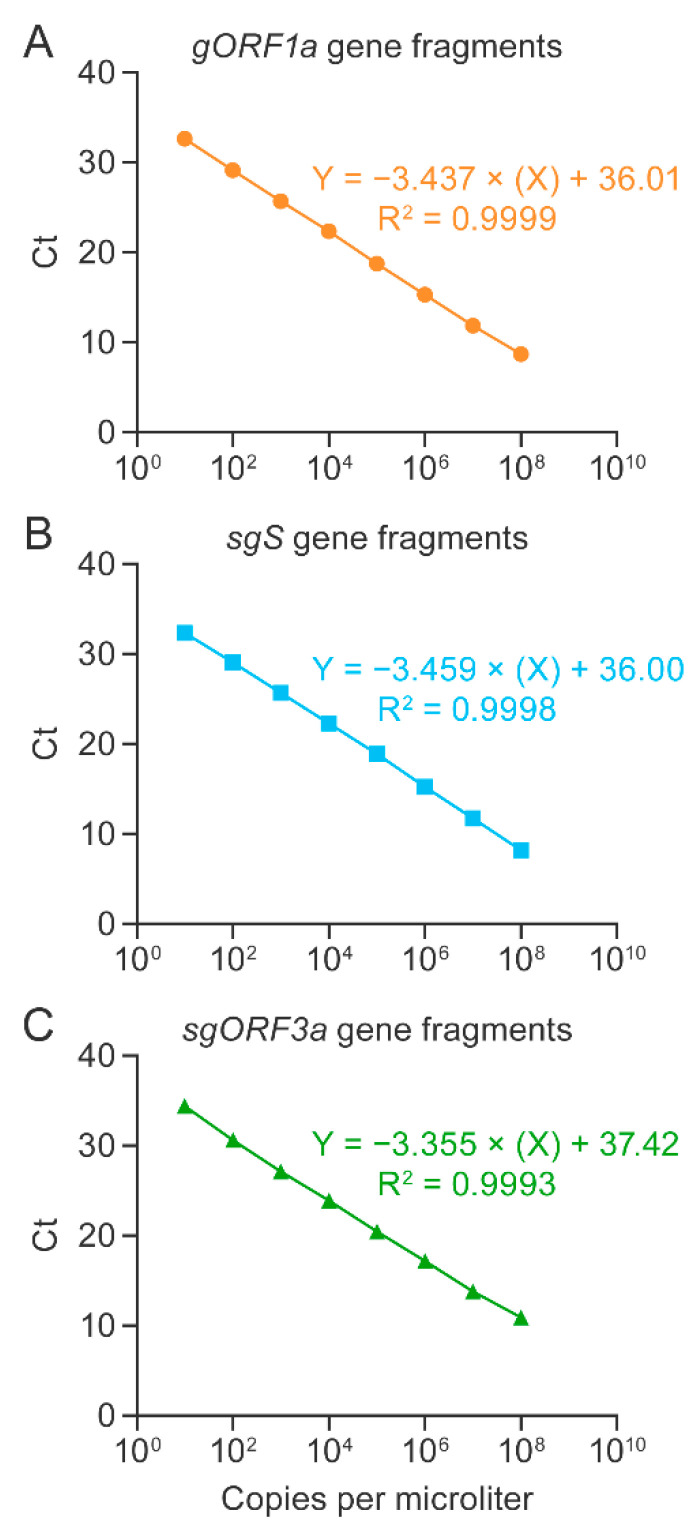
Singleplex RT-qPCR assays for the detection of SARS-CoV-2 RNA using synthetic gene fragments. Singleplex RT-qPCR assays were performed on serially diluted gene fragment standards to optimize detection of SARS-CoV-2 genomic *ORF1a* (*gORF1a*; (**A**)), subgenomic *S* (*sgS*; (**B**)) and subgenomic *ORF3a* (*sgORF3a*; (**C**)). Experiments were performed thrice, with three replicates for each reaction. RT-qPCR = real-time reverse transcription PCR; SARS-CoV-2 = severe acute respiratory syndrome coronavirus 2.

**Figure 2 viruses-14-01066-f002:**
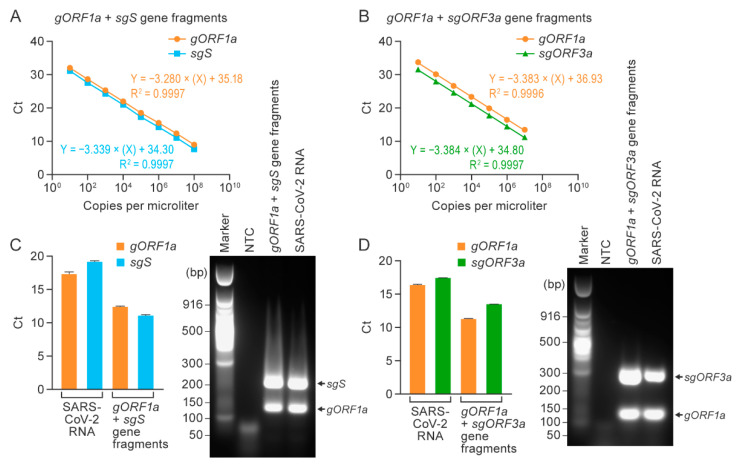
Duplex RT-qPCR assays specifically detect SARS-CoV-2 genomic and subgenomic nucleic acids. Duplex RT-qPCR assays were performed on serially diluted synthetic gene fragment mixtures using (*gORF1a + sgS*; (**A**)) or (*gORF1a + sgORF3a*; (**B**)). Experiments were performed thrice, with three replicates for each reaction. Mean Ct values for duplex assays for RNA from SARS-CoV-2-exposed Vero cells and positive controls (*gORF1a + sgS/gORF1a + sgORF3a*) gene fragments (**C**,**D**). Error bars depict standards deviations. Gel images show amplification products after RT-qPCR in (**C**,**D**). RT-qPCR = real-time reverse transcription PCR; SARS-CoV-2 = severe acute respiratory syndrome coronavirus 2.

**Figure 3 viruses-14-01066-f003:**
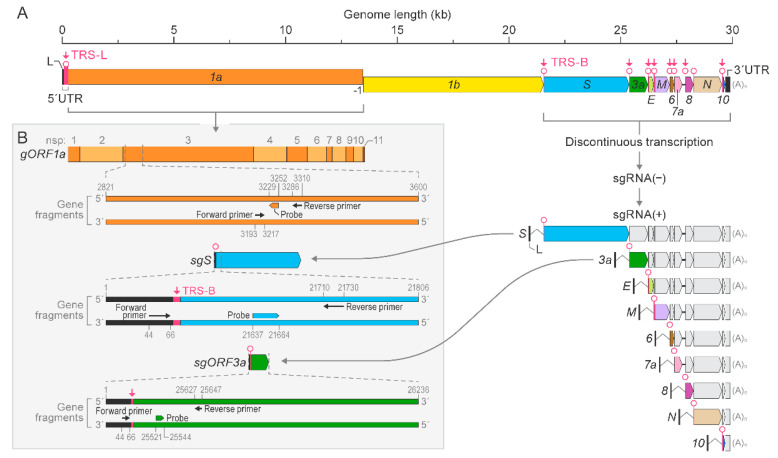
Schematic of SARS-CoV-2 genome organization and transcription. SARS-CoV-2 genomic RNA (*gORF1a* and *gORF1b*) and subgenomic RNA (*sgRNA*) structure (comprising *S*, spike; *E*, envelope; *M*, membrane; and *N*, nucleocapsid) with 5′ leader sequence and transcriptional regulatory sequences (TRS-L and TRS-B) (**A**). The negative-sense and positive-sense *sgRNA*—sgRNA (−) and sgRNA (+), respectively—arising from discontinuous transcription and the primer and probe binding regions for *gORF1a*, *sgS*, and *sgORF3a* gene fragments are illustrated (**B**). SARS-CoV-2 = severe acute respiratory syndrome coronavirus 2.

**Figure 4 viruses-14-01066-f004:**
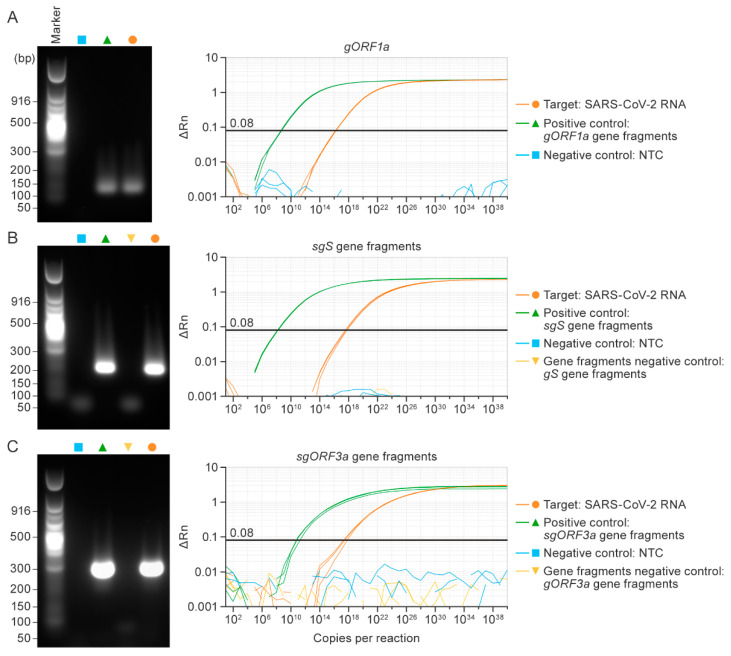
Singleplex RT-qPCR assays specifically detect SARS-CoV-2 genomic or subgenomic nucleic acids. The amplification product after RT-qPCR for genomic *ORF1a* (*gORF1a*; (**A**)), subgenomic spike (*sgS*; (**B**)), and subgenomic *ORF3a* (*sgORF3a*; (**C**)). SARS-CoV-2 RNA from supernatants of infected Vero cells (orange) and positive control gene fragment (green), negative non-template control (NTC; blue). The genomic *S* (*gS*; (**B**)), genomic *ORF3a* (*gORF3a*; (**C**)) were used as negative control gene fragments (gold). Amplification curves for the respective product seen on the gel for target *gORF* (**A**), *sgS* (**B**), and *sgORF3a* (**C**) are shown. The experiment was performed thrice, and each reaction was run in triplicate. RT-qPCR = real-time reverse transcription PCR; SARS-CoV-2 = severe acute respiratory syndrome coronavirus 2.

**Figure 5 viruses-14-01066-f005:**
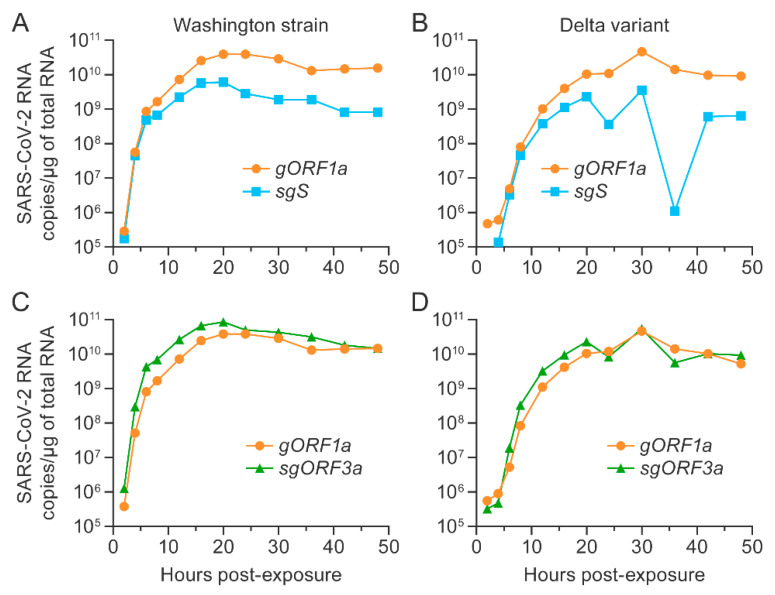
Longitudinal detection of SARS-CoV-2 genomic and subgenomic RNAs in Vero E6 cells expressing TMPRSS2. SARS-CoV-2 genomic and subgenomic RNAs were detected longitudinally using the *gORF1a + sgS* duplex RT-qPCR assay (**A**,**B**) and the *gORF1a + sgORF3a* duplex RT-qPCR assay (**C**,**D**) in TMPRSS2 Vero E6 cells that had been exposed to SARS-CoV-2. Experiments were performed thrice in 6-well plates, and duplex assays were performed in triplicates for each well. SARS-CoV-2 = severe acute respiratory syndrome coronavirus 2; RT-qPCR = real-time reverse transcription PCR; TMPRSS2 = human transmembrane serine protease 2.

**Figure 6 viruses-14-01066-f006:**
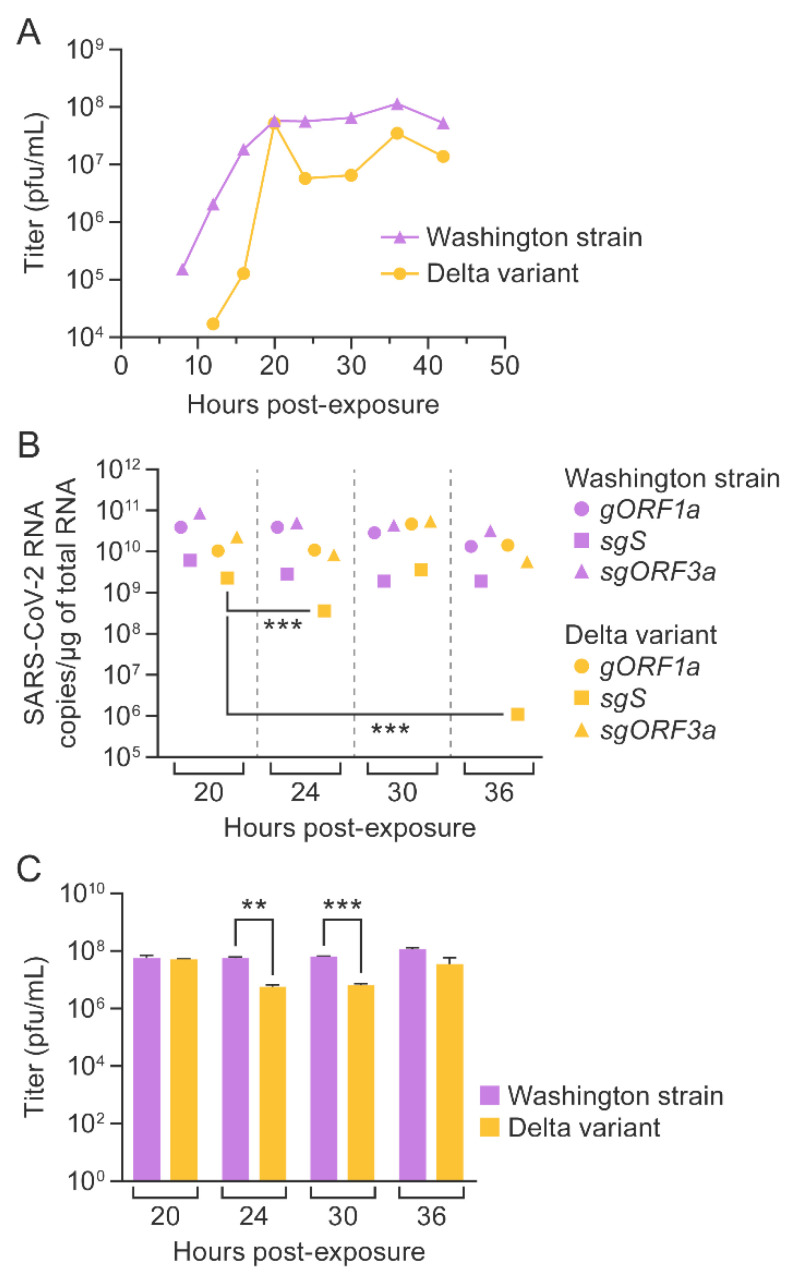
SARS-CoV-2 Washington strain and delta variant titers correlate with genomic or subgenomic RNA copy numbers. Plaque assays were performed for supernatants collected from human Vero E6 cells expressing TMPRSS2 exposed to SARS-CoV-2 (MOI = 0.01) at various time points (**A**). Experiments were performed thrice, and plaque assays were performed in duplicate for each infection. Comparison of SARS-CoV-2 genomic and subgenomic RNA copies in TMPRSS2 Vero E6 cells at 20, 24, 30, and 36 h after exposure to SARS-CoV-2 Washington strain or delta variant. Statistical significance is represented as *** (*p* = 0.001) on the scatter plot (**B**). Average SARS-CoV-2 titers from three independent experiments during which infected TMPRSS2 Vero E6 supernatants were collected for plaque assays at 20, 24, 30, and 36 h post-exposure (**C**). Statistical significance, established with an unpaired two-tailed t-test, is presented as ** (*p* < 0.05) and *** (*p* < 0.001). Error bars represent standard deviations. SARS-CoV-2 = severe acute respiratory syndrome coronavirus 2; RT-qPCR = real-time reverse transcription PCR; TMPRSS2 = human transmembrane serine protease 2.

**Table 1 viruses-14-01066-t001:** Primer and probe sets for detection of SARS-CoV-2 genomic and subgenomic RNAs.

Target Regions	Primers and Probes	Sequences (5′–3′)
Genomic *ORF1a*	*gORF1a*-F *gORF1a*-R *gORF1a*-P	AGAAGATTGGTTAGATGATGATAGT TTCCATCTCTAATTGAGGTTGAACC *FAM*/TCCTCACTGCCGTCTTGTTGACCA/*BHQ13*
Subgenomic *S*	sgS-F sgS-R *sgS*-P	CGATCTCTTGTAGATCTGTTCTC TAAGAACAAGTCCTGAGTTGA *SUN*/CCCTGCATA/*ZEN*/CACTAATTCTTTCACACGT/*IABKFQ*
Subgenomic *ORF3a*	sgORF3a-F sgORF3a-R *sgORF3a*-P	CGATCTCTTGTAGATCTGTTCTC CAACAGCAAGTTGCAAACAAA *ABY*/CGGATGGCTTATTGTTGGCGTTGC/*QSY7*

SARS-CoV-2 = severe acute respiratory syndrome coronavirus 2; g = genomic; sg = subgenomic; F = forward primer; P = probe; R = reverse primer.

**Table 2 viruses-14-01066-t002:** Ct values for singleplex and duplex RT-qPCR assays using supernatant of Vero cells infected with SARS-CoV-2 Washington strain.

Assay	Target Regions	Ct Mean	Ct SD
Singleplex	Genomic *ORF1a*	16.620	0.033
	Subgenomic *S*	17.669	0.130
	Subgenomic *ORF3a*	17.517	0.212
Duplex *ORF1A/sgS*	Genomic *ORF1a*	17.397	0.213
	Subgenomic *S*	19.160	0.184
Duplex *ORF1a/sgORF3a*	Genomic *ORF1a*	16.981	0.181
	Subgenomic *ORF3a*	18.399	0.103

RT-qPCR = real-time reverse transcription PCR; SARS-CoV-2 = severe acute respiratory syndrome coronavirus 2; sg = subgenomic.

## Data Availability

The data presented in this study are available in Appendix A.

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
