# Peer review of "Duplex One-Step RT-qPCR Assays for Simultaneous Detection of Genomic and Subgenomic RNAs of SARS-CoV-2 Variants"

_viruses, 2022, doi:10.3390/v14051066_

Round 1
Reviewer 1 Report
The authors developed a duplex one-step RT-qPCR assays for the simultaneous detection of genomic and subgenomic RNAs of SARS-CoV-2. The manuscript is well-written, and the experiments are well-designed.
I have a few suggestions/queries:
- The authors mentioned that the primers and probes were designed to detect SARS-CoV-2 variants including omicron. However, the experiments were done only with Washington strain and delta variant. It would be interesting to check the assay with more variants to prove its wider usefulness.
- On line 22, you may want to change the ‘in this experiment’ to ‘in this paper/study’.
- On line 100, it is mentioned that TMPRSS Vero E6 cells were cultured in MEM with 2% FBS. However, on line 109, it is mentioned that TMPRSS cells were cultured in DMEM containing 10 % FBS and 1 mg/mL of G418. Are these for different sets of experiments?
- On line 222, it states that ’34 h’, is it ’24 h’? Please check once.
- Figure 2 a has 8 points and figure 2b has 7 points, is there any specific reason? To be consistent, both graphs can have the same number of points, if possible.
- It would be very interesting to know how the levels of sgS change during active replication/non-replication of the virus. Like comparing the various Ct value samples or culturable/non-culturable samples to check levels of sgS using the duplex assay. It will emphasize the importance of the assay in the diagnostics settings.
Author Response
The authors developed a duplex one-step RT-qPCR assays for the simultaneous detection of genomic and subgenomic RNAs of SARS-CoV-2. The manuscript is well-written, and the experiments are well-designed.
Response: We thank the reviewer for the positive assessment of our work.
I have a few suggestions/queries:
- The authors mentioned that the primers and probes were designed to detect SARS-CoV-2 variants including omicron. However, the experiments were done only with Washington strain and delta variant. It would be interesting to check the assay with more variants to prove its wider usefulness.
Response: The reviewer raises an important point. We indeed used this assay by now to detect other variants, including lambda and omicron in vitro and have utilized this assay to detect omicron isolates collected from nonhuman primate samples. Unfortunately, we are unable to include these data, as they are part of other projects and are beyond the scope of the current manuscript. Without these data in the manuscript, we agree with the reviewer that although the alignment indicate that the omicron variant would be covered by our assay, it is best to delete the mentioning of omicron from the abstract.
- On line 22, you may want to change the ‘in this experiment’ to ‘in this paper/study’.
Response: We have made this change as suggested.
- On line 100, it is mentioned that TMPRSS Vero E6 cells were cultured in MEM with 2% FBS. However, on line 109, it is mentioned that TMPRSS cells were cultured in DMEM containing 10 % FBS and 1 mg/mL of G418. Are these for different sets of experiments?
Response: We have modified the text to make it clearer, moving the statement from line 109 up to merge with line 100.
- On line 222, it states that ’34 h’, is it ’24 h’? Please check once.
Response: We confirm that 34 h is correct on line 222.
- Figure 2 a has 8 points and figure 2b has 7 points, is there any specific reason? To be consistent, both graphs can have the same number of points, if possible.
Response: Thank you for this suggestion. Although we generally agree with you, the scales of the axes were selected for the following reason: In Figure 2a, the highest concentration of standards was 1x108 copies per µL used for the ORF1a/sgS duplex assay. However, for the ORF1a/ORF3a assay, this concentration was too high and out of range, which is why it was not graphed. At this concentration, one target out of the two in the duplex assay was inhibited and the amplification curve was not normal, so we have 7 points, starting with 1x107 copies per µL.
- It would be very interesting to know how the levels of sgS change during active replication/non-replication of the virus. Like comparing the various Ct value samples or culturable/non-culturable samples to check levels of sgS using the duplex assay. It will emphasize the importance of the assay in the diagnostics settings.
Response: We agree with your comment. In Figure 5, we show how the sgS changes longitudinally when the virus is replicating actively in TMPRSS2 Vero cells. In the same experiment, we also collected the samples for plaque assay at various time points post‑infection, showing active replication over time. As mentioned in our reply to comment #1, for some other studies we used this assay on samples collected several weeks after exposure of nonhuman primates when the virus was not replicating. Although we may not include the data here, we confirm that we did not detect any sgS in samples that did not contain infectious SARS-CoV-2.

Reviewer 2 Report
he manuscript (manuscript ID: viruses-1695601) entitled “Duplex one-step RT-qPCR assays for simultaneous detection of genomic and subgenomic RNAs of SARS-CoV-2 variants” by Dr. Bhosle reports on a duplex one-step RT-qPCR methods capable in detecting SARS-CoV-2 genomic ORF1a and and either subgenomic spike or subgenomic ORF3a RNAs. The method successfully detected SARS-CoV-2 Washington strain and delta variant isolates at various time points during the course of live virus infection in vitro. Despite several sections, such as the methods, are too verbose and difficult to read, the present manuscript is well written. The experimental design is well performed. Figures and tables are highly informative. The novel RT-qPCR-based method described in the present manuscript will improve our knowledge behind the SARS-CoV-2 detection methods. Taking into consideration the aforementioned aspects, I therefore recommend a minor revision. I have few suggestions for improving the manuscript.
Thank you for the opportunity to review this work.
General comments
1. The main weakness of the study is the lack of validation with clinical samples from SARS-CoV-2-affected patients/individuals. This important limitation should be included in the discussion.
2. The methods section is almost completely lacking in supporting references. Reference should be included. Moreover, if not mandatory, products' catalogue numbers should be removed (or moved to supplemental) for a better readability
3. Figure 2, similarly to panels C and D, the color legends should be detailed also in panels A and B
4. I suggest removing the figure quotes throughout the discussion section
Minor
Lines 37-48 These sentences are lacing in supporting references. A detailed description of the SARS-CoV-2 genome and proteins is reported here (DOI: 10.3390/v13091687) and here (doi: 10.1016/j.ygeno.2020.09.059).
Lines 46-47 Considering their importance, SARS-CoV-2 protein’s names should be moved from the parenthesis
Lines 259-276 The sentences would better fit in the discussion. I suggest rephrasing this paragraph, by maintaining the essential information in the results and moving this entire part in the discussion
Line 282 Were the standard curves performed in triplicate/duplicate? If yes this information should be included
Line 467 For completeness of information, besides the ddPCR approach for detecting SARS-CoV-2 RNA, additional, more recent methods such as Next generation sequencing (doi: 10.1093/bib/bbaa297) and the CRISPR–Cas12-based assay (doi: 10.1038/s41587-020-0513-4.) should be mentioned. It would be helpful for the reader
Author Response
he manuscript (manuscript ID: viruses-1695601) entitled “Duplex one-step RT-qPCR assays for simultaneous detection of genomic and subgenomic RNAs of SARS-CoV-2 variants” by Dr. Bhosle reports on a duplex one-step RT-qPCR methods capable in detecting SARS-CoV-2 genomic ORF1a and and either subgenomic spike or subgenomic ORF3a RNAs. The method successfully detected SARS-CoV-2 Washington strain and delta variant isolates at various time points during the course of live virus infection in vitro. Despite several sections, such as the methods, are too verbose and difficult to read, the present manuscript is well written. The experimental design is well performed. Figures and tables are highly informative. The novel RT-qPCR-based method described in the present manuscript will improve our knowledge behind the SARS-CoV-2 detection methods. Taking into consideration the aforementioned aspects, I therefore recommend a minor revision. I have few suggestions for improving the manuscript.
Response: We thank the reviewer for the positive assessment of our work.
Thank you for the opportunity to review this work.
General comments
1. The main weakness of the study is the lack of validation with clinical samples from SARS-CoV-2-affected patients/individuals. This important limitation should be included in the discussion.
Response: We have included a statement on the lack of validation in clinical samples in the discussion (lines 468-470). The assay is currently being utilized for detection of SARS-CoV-2 in samples collected from nonhuman primates at different stages of infection.
The methods section is almost completely lacking in supporting references. Reference should be included. Moreover, if not mandatory, products' catalogue numbers should be removed (or moved to supplemental) for a better readability.
Response: Thank you; we agree with you. We have removed the reagent catalogue numbers, moved text in section 2.1- cells and 2.3- RNA extractions for better readability. As suggested, we have also added lacking references to the plaque assay (Reference 38) and RNA extraction methods (References 40 and 41).
Figure 2, similarly to panels C and D, the color legends should be detailed also in panels A and B
Response: We are glad to add the color legends to panel A and B similar to panels C and D for Figure 2.
I suggest removing the figure quotes throughout the discussion section
Response: We have removed all figure quotes throughout the discussion section.
Minor
Lines 37-48 These sentences are lacing in supporting references. A detailed description of the SARS-CoV-2 genome and proteins is reported here (DOI: 10.3390/v13091687) and here (doi: 10.1016/j.ygeno.2020.09.059).
Response: References (DOI: 10.3390/v13091687 and DOI: 10.1016/j.ygeno.2020.09.059) suggested for lines 37-48 have been added.
Lines 46-47 Considering their importance, SARS-CoV-2 protein’s names should be moved from the parenthesis
Response: The protein names were moved from the parenthesis.
Lines 259-276 The sentences would better fit in the discussion. I suggest rephrasing this paragraph, by maintaining the essential information in the results and moving this entire part in the discussion
Response: Thank you for this suggestion. Lines 259-276 were moved to the discussion except the sentence “To develop a duplex RT-qPCR assay for detection of SARS-CoV-2 genomic and subgenomic nucleic acids, we first established and optimized singleplex RT-qPCR assays targeting gORF1a, sgS, or sgORF3a sequences”, which fits well in the results. The lines moved to the discussion are lines 411-414 and lines 416-428 in the revised version.
Line 282 Were the standard curves performed in triplicate/duplicate? If yes this information should be included
Response: Standard curves were run in triplicate. This is mentioned in the figure legends for Figure 1 and Figure 2. We have also included it in the results text (Line 282), as suggested.
Line 467 For completeness of information, besides the ddPCR approach for detecting SARS-CoV-2 RNA, additional, more recent methods such as Next generation sequencing (doi: 10.1093/bib/bbaa297) and the CRISPR–Cas12-based assay (doi: 10.1038/s41587-020-0513-4.) should be mentioned. It would be helpful for the reader
Response: Thank you for your helpful suggestions. On line 467, we added references for sequencing and CRISPR-CAS12 methods; they are now part of the references list, as well.
